# Ensiling Characteristics, In Vitro Rumen Fermentation Patterns, Feed Degradability, and Methane and Ammonia Production of Berseem (*Trifolium alexandrinum* L.) Co-Ensiled with Artichoke Bracts (*Cynara cardunculus* L.)

**DOI:** 10.3390/ani13091543

**Published:** 2023-05-04

**Authors:** Mariam G. Ahmed, Adham A. Al-Sagheer, Ahmed M. El-Waziry, Samir Z. El-Zarkouny, Eman A. Elwakeel

**Affiliations:** 1Department of Animal and Fish Production, Faculty of Agriculture (El-Shatby), Alexandria University, Alexandria 21545, Egypt; mgmal1984@yahoo.com (M.G.A.); aelwaziry@yahoo.com (A.M.E.-W.); selzarkouny@alexu.edu.eg (S.Z.E.-Z.); emankeel@yahoo.com (E.A.E.); 2Department of Animal Production, Faculty of Agriculture, Zagazig University, Zagazig 44511, Egypt

**Keywords:** legumes forages, artichoke bracts, silage fermentation quality, methane, ammonia, in vitro rumen fermentation

## Abstract

**Simple Summary:**

Despite the importance of legume forage silage in ruminant livestock feeding, its low water-soluble carbohydrate content, great buffering capacity, and urea release to the environment limit its value. Ensilage of artichoke byproducts appears to be an environmentally efficient means of disposing of artichoke crops waste. Hence, this study aimed to investigate the effect of berseem co-ensiling with graded levels of artichoke bracts on silage characteristics. Moreover, the changes in ruminal fermentation characteristics and methane and ammonia production were evaluated using a buffalo inoculum source. The results showed that the co-silage of berseem and artichoke bracts considerably enhanced the silage quality, particularly after 30 days of ensiling at intermediate ratios. Moreover, the in vitro rumen degradation was significantly improved by artichoke bracts concentration of 500 g/kg fresh forage.

**Abstract:**

This study investigated the effect of co-ensiling increasing levels of artichoke bracts (*Cynara cardunculus* L.) with berseem (*Trifolium alexandrinum* L.) (100:0, 75:25, 50:50, 25:75, and 0:100, respectively) on silage quality after 0, 30, 60, and 120 days. Moreover, the in vitro rumen fermentation characteristics and methane (CH_4_) and ammonia (NH_3_-N) production were evaluated using a buffalo inoculum source. The results showed that pH of the silage and the concentration of acetic, propionic, butyric acid, and NH_3_-N significantly decreased (L; *p* < 0.01) with the increasing amounts of artichoke bracts in the mixture. At 30 and 60 days of ensiling, the highest lactic acid concentration was observed at intermediate proportions of artichoke bracts (*p* < 0.01). Cumulative gas production was higher in artichoke bracts than in the berseem silage. After 24 h of incubation, the highest value (*p* < 0.05) of truly dry matter, organic matter, natural detergent fiber degradability, and NH_3_-N concentration was recorded with 500 g/kg of forage mixtures. As the artichoke bract concentration increased, the partitioning factor and ruminal pH declined linearly (*p* ≤ 0.05). No significant differences were observed for total volatile fatty acids and volatile fatty acids molar proportions. In summary, co-ensiling artichoke bracts with berseem at a ratio of 1:1 might be a promising and easy method for the production of high-quality silage from legume forage with positively manipulating rumen fermentation.

## 1. Introduction

Legume forages such as alfalfa and berseem are considered excellent forage sources for ruminant livestock because of their high protein, calcium, and high-quality fiber content [1]. Berseem, or *Trifolium alexandrinum* L., is a fast growing, high-quality leguminous forage widely cultivated in the Middle East and Mediterranean regions [2]. This species has an advantage over other annual species, providing multiple harvests throughout the growing season [3]. It is frequently compared to alfalfa because of its similar nutritive value. Yet, unlike alfalfa, it has lower bloating potential [4]. In particular, the silage of legumes, compared to dried herbs, has a higher feeding value. Nevertheless, ensiling legume forages are less successful than ensiling cereal crops because of their comparatively low water-soluble carbohydrate content and strong buffering capacity [5]. Moreover, from the environmental perspective, the excessive protein degradation of legume forages during ensiling prevents the reduced pH of silage and dissipates more nitrogen as urea in urine, which has a negative environmental impact [6]. Hence, there is a growing global interest in modifying the ensiling strategies of legume forages to enhance the silage quality, maintain its nutritional value, and mitigate its environmental impact [7,8]. Several attempts have been made to enhance the ensiling capabilities of legume forages via additives, including acidifiers, bacterial inoculants, tannic acid, and formaldehyde. Still, the results of their usefulness are conflicting [9,10]. However, some chemicals are difficult to handle because of their pungent and offensive odor [10]. Therefore, several researchers have been directed to use alternatives, such as phytogenic substances, to improve silage quality [11]. For instance, some studies demonstrated that the plant byproducts high in phytogenic substances could modify the silage fermentation by reducing proteolysis, butyric acid production, loss of dry matter (DM), and growth of underside microbes (e.g., Clostridia and Enterobacter) and increasing lactic acid production [12,13,14,15].

Artichoke (*Cynara cardunculus* L.) is a member of the *Asteraceae* family that is widely cultivated in Mediterranean regions such as Italy, Egypt, and Spain [16]. A total of 1.52 Mt of globe artichokes were harvested worldwide in 2020, making it one of the most widely grown vegetables [17]. In Egypt, annual artichoke production represents 296,899 tons [18]. Over 80% of an artichoke’s biomass comprises non-edible parts such as leaves, outer bracts, and stems [19,20]. These byproducts are high in minerals, vitamin C, water-soluble polysaccharides (e.g., inulin), and active components such as triterpenes, sesquiterpenes, and phenolic compounds (e.g., flavonoids, caffeoylquinic acids, and anthocyanins) [16,21]. Using artichoke byproducts has recently been included for biofuel production and forage for animals to minimize waste discharges and waste management costs [22,23]. Moreover, previous fermentative parameters studies have demonstrated that artichoke byproducts had a high propensity for ensilage [24,25]. In the earlier studies, no phytosanitary products were found after 12 days of ensilage in artichoke byproducts silages. Additionally, the authors argued that artichoke byproducts silages were harmless and had sufficient nutritional value to be used as feeds for ruminants, making them a sustainable option for removing wastes from artichoke crops processing.

To our knowledge, no available studies have examined the effect of co-ensiling artichoke with berseem. Hence, based on the aforementioned favorable characteristic of artichoke byproducts, we hypothesized that co-ensiling artichoke bracts with one of the legume forages, berseem could produce legume forage silage with enhanced quality and favorably manipulate rumen fermentation. To test this hypothesis, this study investigated the effect of diverse levels of artichoke bracts on the fermentation quality of berseem silage at different ensiling times. Moreover, the rumen fermentation characteristics and methane (CH_4_) and ammonia (NH_3_-N) production were assessed in vitro using a buffalo bulls inoculum source.

## 2. Materials and Methods

The current study was conducted at the Laboratory of Animal Nutrition, Department of Animal and Fish Production, Faculty of Agriculture (El-Shatby), Alexandria University. 

### 2.1. Forage Materials and Ensiling

Berseem samples were obtained from a private commercial farm in the Abees region Alexandria, which uses an irrigation system in which it was harvested as a 4th cut. The artichoke was also collected from a private farm in the Abees, Alexandria. The external bracts of the artichoke were separated manually from the edible parts of the artichoke. Afterward, forage materials were wilted for 24 h and manually chopped into 1.5–2 cm pieces by a hand cutter. Berseem and artichoke bracts were mixed manually at ratios of 100:0 (A0), 75:25 (A25), 50:50 (A50), 25:75 (A75), and 0:100 (A100, wt:wt on a fresh weight basis), respectively. The forage materials were then compacted and sealed with a plastic lid and covered with duct tape into a laboratory scale mini plastic silo (16 cm height, ×9 cm diameter) with a 1 kg capacity. Sixty plastic silos (5 treatments × 4 sampling times × 3 replicates) were stored at room temperature (20–22 °C) and opened after 0, 30, 60, and 120 days.

### 2.2. Silage Fermentation Characteristics

At the end of each ensiling time, the content of each silo from each treatment was individually evaluated. The temperature was recorded by inserting a thermometer into the plant mass center according to Meneses et al. [25]. The content of each silo from the treatment was mixed thoroughly and divided into two sub-samples. The first subsample from each replicate was taken for assaying water activity (a_w_) using the Decagon Aqualab CX-1 (Decagon Devices, Inc., Pullman, WA, USA). Additionally, 20 g of samples in duplicates were homogenized for 1 min in 100 mL distilled water and kept at room temperature for 1 h. Four cheesecloth layers were used to filter the water extract according to Madrid et al. [26], and then pH was measured using a pH meter (Adwa AD 11 waterproof; Szeged-Hungary Europe, Romania). One milliliter of filtered liquid was centrifuged at 30,000× *g* for 20 min. The supernatant was transferred to a 1.5 mL Eppendorf tube for lactic acid concentration determination using iron (III) chloride at 390 nm, as described by Borshchevskaya et al. [27]. To determine ethanol, NH_3_-N, and volatile fatty acids (VFA) concentration, 1 mL of filtered liquid was mixed with 200 μL meta-phosphoric acid 25% (*w*/*v*) into a 1.5 mL Eppendorf tube and kept at −20 °C for analysis. Samples were centrifuged for 20 min at 30,000× *g*. Then, the supernatant was transported to vials [28]. The ethanol and VFA were measured via gas chromatography (GC Thermo TRACE 1300) using a capillary column (30 m TR-FFAP × 0.53 mmI D × 0.5 μm film (thermo-part NO: 260 N225 P). The temperature was 100–200 °C at a 10 °C/min rate. The injection and flame ionization detector (FID) temperatures were set at 220 °C and 250 °C, respectively. The carrier gas, nitrogen, and flow rate were adjusted to 7 mL/min. There was a 450 mL/min gas flow, a 40 mL/min hydrogen flow, and a 35 mL/min make-up gas flow. Calibration was performed using a standard of VFA concentrations rather than an internal standard. A commercial kit produced by Biodiagnostic Company, Egypt, was used to assess NH_3_-N concentration colorimetrically. The second subsample was taken to determine DM via 72 h drying at 50 °C in an air-forced oven, then ground using a 1mm screen. The organic matter (OM), ash, and crude protein (CP; N × 6.25) were determined [29]. Neutral detergent fiber (NDF), acid detergent lignin (ADL), and acid detergent fiber (ADF) were measured by an ANKOM 220 fiber analyzer (ANKOM, model A2001, Macedon, NY, USA) in line with the protocol of Van Soest et al. [30]. No heat-stable amylase or sodium sulfite was used to analyze the NDF. Hemicellulose was determined by subtracting NDF from ADF and cellulose from ADF minus ADL.

### 2.3. In Vitro Incubation

The ruminal inoculum was obtained from three slaughtered buffalo bulls (500 ± 25 kg, body weight) at the slaughterhouse of the Agricultural Experimental Station of the Faculty of Agriculture (El-Shatby), Alexandria University, Alexandria, Egypt. Ruminal content collection from slaughtered animals reduces stress imposed on live animals requiring a surgical cannula with full compliance with animal welfare regulations [31,32]. These animals were fed on commercial concentrate (140 g/kg of CP) and rice straw as a basal diet. Rumen fluid was collected separately from each animal immediately after slaughter in a pre-warm insulated flask (39 °C) and conveyed under anaerobic settings to the laboratory within 30 min. In the laboratory, ruminal fluid was pooled into equal portions from each animal in the beaker and strained using four layers of cheesecloth, then placed on the magnetic stirrer at 39 °C and flushed with CO_2_. Before the day experiment, McDougall’s buffer was prepared according to McDougall [30], and 500 mg of silage samples were weighed in triplicates into a serum bottle (120 mL), then placed in incubators at 39 °C. On the day of the experiment, ruminal fluid and buffer solution were dispensed at a ratio of 1:2 *v*/*v* into each bottle and flushed with CO_2_, then closed with a rubber stopper and aluminum rumples. A23 G needle was inserted into the bottle’s stopper to adjust the pressure in the headspace. The bottles were then placed in the incubator and heated to 39 °C. Buffered rumen fluid was used to fill serum bottles in triplicate without a substrate to serve as a blank. 

An in vitro gas production technique using a semi-automated system was used for the evaluation of artichoke bracts that were ensiled with berseem (A0, A25, A50, A75, and A100) for 30 days of ensiling periods. Approximately 200 g of silage samples from each mini-silo of each treatment were thoroughly mixed. Subsamples were taken and dried by an air-forced oven at 50 °C for 72 h and then ground to 1mm for using a feed substrate.

### 2.4. Sample Collection and Measurements

To prevent pressure reading errors, incubator bottles were transferred to a water bath maintained at 39 °C while gas pressure was recorded [33]. According to Mauricio et al. [32], cumulative gas production was measured using a data logger and pressure transducer (GN200, Sao Paulo, Brazil) at 3, 6, 9, 12, 24, 48, and 72 h of incubation. Following each reading, the 23 G needle was injected to release gas pressure to zero because gas pressure exceeding 48.3 Kpa will negatively impact microbial growth [34]. Depending on our laboratory conditions, it is located at a 76 m altitude, the average atmospheric pressure (psi) at this site is 14.565 psi, and the headspace of the serum bottle (V h) is 70 mL. The gas production volume at every incubation time was calculated in line with the equation described by López et al. [35]: V = 4.8060 × Pt, where V = gas volume (mL); Pt = the pressure estimated using a transducer (Psi). The exponential model of Ørskov and McDonald [36] and the Fit Curve software developed by Chen [37] were used to fit the cumulative gas production (Y) as a function of time (t) to measure gas production kinetics (mL/200 mg DM): Y = a + b × (1 − exp^−ct^)(1)
where a = the amount of gas produced from the soluble fraction (mL), b = the gas produced from the insoluble fraction (mL), c = rate constant for gas produced via insoluble fraction (b), and t = incubation time.

One milliliter of gas was collected via syringe at 3, 6, 9, 12, and 24 h of incubation and the collected representative sample from the five time points (5 mL) was transferred into 5 mL glass vacuum airtight tubes (BD Vacutainer^®^ Tubes, Franklin Lakes, NJ, USA). Then, 1 mL from the collected 5 mL was injected into the GC for CH_4_ analysis [38]. The CH_4_ was measured using gas chromatography (Model 2014, Drawell Scientific Instrument Co., Ltd., Shanghai, China) intended with a Molesieve 5A micro packed column (1 m, 2 mm ID, Ref no. 80440-800; Restek, Bellefonte, PA, USA) and thermal conductivity detector. Helium was used as the carrier gas at a flux of 30 mL/min, and the column, injector, and detector were heated to temperatures of 50 °C, 150 °C, and 200 °C, respectively. A standard gas curve covering the range of sample concentrations was used for linearity and calibration tests. The CH_4_ production was calculated as follows: CH_4_ (mL) = CH_4_ concentration × (total gas (mL) + headspace)(2)

Net CH_4_ and GP were corrected for the corresponding blank values. After 24 h incubation time, serum bottles of each treatment were put on ice to stop fermentation. The liquid phase (1 mL) was carefully collected into a syringe by inserting a 23-gauge needle into the bottle’s rubber stopper and placed in small vials to determine the pH using a pH meter (GLP21 model; CRISON, Barcelona, Spain). An additional 1 mL of the liquid phase was taken and dispensed into a 1.5 mL microtube to measure VFA and NH_3_-N, as defined in Section 2.2. The residues of the bottle were filtered through pre-weighed crucibles and washed with distilled water. The crucibles were dried at 110 °C for 24 h to determine the apparent dry matter degradation, which was calculated by subtracting full crucibles from empty crucibles and correcting for blank residue. The true degradability was determined as described by López et al. [35]. The dry residue was collected and weighed in pre-weighed filter bags (F57-ANKOM Technology Corporation, Macedon, NY, USA) and then extracted with neutral detergent solution NDS at 100 °C for 1 h in an ANKOM fiber analyzer. The bags were dried and weighed overnight at 100 °C before being transferred to a muffle furnace for 2 h at 600 °C to determine the true organic matter degradation. By subtracting the weight of the substrate after NDS solution extraction from the weight of the substrate before incubation, the substrate’s true DM and NDF degradability was determined. The partition factor (PF) was calculated as the ratio of digested dry matter (mg) to gas volume (mL) consistent with Blümmel et al. [39]. Microbial protein (MP) was calculated as 19.3 g of microbial nitrogen/OMD kg along with the equation of Czerkawski [40]. 

### 2.5. Statistical Analyses

All data analysis was performed using the Mixed procedure of SAS (version 9.0, SAS Institute Inc., Cary, NC, USA) with the fixed effects of forage type, ensiling time, and their interaction included in the model. Orthogonal contrast was conducted to define the linear and quadratic effects of the means. Tukey’s test was adopted to compare the differences between the forage means, and the significance was declared at *p* < 0.05.

## 3. Results

### 3.1. Silage Characteristics 

#### 3.1.1. Physical Properties

The physical properties of fresh and ensiled forages are presented in Table 1. No significant alterations were observed between forage type and forage type × ensilage time interaction on temperature and water activity (a_w_). Instead, the ensiling time significantly increased (*p* < 0.05) temperature and a_w_ for all ensiled forages. A significant effect (*p* < 0.01) of forage type, ensiling time, and forage type × ensiling time interaction was observed based on the pH of forages. With increased artichoke ratios in the silage mixture, the pH decrease was quadratic and cubic (*p* < 0.01) at day 0, linear (*p* < 0.01) and quadratic (*p* < 0.05) at day 30, linear (*p* < 0.01), quadratic (*p* < 0.05) and cubic (*p* < 0.01) at day 60, and linear (*p* < 0.05) and cubic (*p* < 0.01) at day 120. 

#### 3.1.2. Lactic Acid and VFA

The concentration of lactic acid was also influenced (*p* < 0.05) by forage type, ensiling time, and forage type x ensiling time interaction (Table 2). A maximum concentration of lactic acid was detected during the first 30 days of ensiling for all ensiled forages except for A75 and A100 mixtures. Additionally, the lactic acid concentration responded quadratically (*p* < 0.01) to increasing artichoke bract content in the forage mixture at 30 and 60 days of ensiling. The highest values of lactic acid were found at intermediate mixing ratios. Significant effects of forage type, ensiling time, and their interaction were recorded for the concentration of VFA (*p* < 0.05). The concentration of acetic, propionic, and butyric acid linearly decreased (*p* < 0.01) with increasing artichoke bracts content in the forage mixture, and lower values were observed at 30 days of ensiling. A higher propionic acid concentration was shown with berseem silage (A0) and 250 g/kg fresh forage (A25) mixtures at 60 and 120 days of ensiling. Ethanol was affected by ensiling time and the interaction of forage type and ensiling time (*p* < 0.05). The highest ethanol concentration was observed at 60 days of ensiling for all ensiled forages. All forage mixtures increased the ethanol concentration except 250 g/kg fresh mixture of forages (A25), which decreased ethanol at 60 and 120 days of ensiling.

#### 3.1.3. NH_3_-N and Chemical Composition

The concentration of NH_3_-N in forage mixtures was significantly affected (*p* < 0.05) by forage type, ensiling time, and their interaction (Figure 1). The concentration of NH_3_-N linearly decreased (*p* < 0.01) with increasing amounts of artichoke bracts in the mixture and significantly increased with the increase in the ensiling time. At 30 and 60 days of ensiling, the concentration of NH_3_-N responded quadratically to increasing artichoke bracts content in the mixture (i.e., minimum values at intermediate artichoke bracts concentration (Q: *p* < 0.01).

The effect of forage type, ensiling time, and forage type and ensiling time interaction was significant (*p* < 0.05) for the content of DM, OM, CP, and structural carbohydrates in either fresh or ensiled forages (Table 3). Except for day 0, the content of DM linearly and cubically increased (*p* < 0.01) with the increasing proportion of artichoke bracts in the forage mixtures. Additionally, the OM and NDF content linearly and cubically increased (*p* < 0.01) with the increasing proportion of artichoke bracts in the forage mixtures. On the contrary, at 0 and 30 days of ensiling, CP decreased (*p* < 0.01) in linear, quadratic, and cubic manner with increasing proportion of artichoke bracts in the forage mixtures. An increase in the ratios of artichoke bracts in the mixture either before or after ensiling linearly, quadratically, and cubically increased (*p* < 0.01) the content of structural carbohydrates, including cellulose and hemicellulose. In contrast, ADL content decreased (*p* < 0.01) in linear manner with increasing proportion of artichoke bracts in the forage mixtures. As ensiling time progressed, the content of ensiled DM, OM, and CP forages significantly decreased (*p* < 0.05). The content of structural carbohydrates in ensiled forages significantly increased (*p* < 0.05) with the ensiling time. 

### 3.2. Ruminal Gas Production 

The cumulative gas production for silage of berseem, artichoke bracts, and their mixtures is demonstrated in Table 4. Both forage type and incubation time influenced (*p* < 0.001) cumulative gas production. The production of gas increased from 12 to 72 h with all forage types, and the highest (*p* < 0.05) value of gas production was detected when artichoke bracts were mixed with berseem at 500 g/kg of forage (A50). The silage of artichoke bracts (A100) significantly (*p* < 0.05) amplified gas production compared to the silage of berseem (A0) (101.76 vs. 92.40 mL/g DM). No interaction effects on gas production were found (*p* = 0.2609) between forage type and incubation time.

The parameters of gas production estimated using an exponential model are given in Table 5. The gas produced from the insoluble fraction (b) responded linearly (*p* = 0.03) to increasing artichoke bracts in the mixture. The highest value of the gas production rate (c) (0.072 mL/h) was found with 500 g/kg of forage, and the lowest rate (0.041 mL/h) was observed with 750 g/kg of fresh forage. Additionally, no significant differences were found between forages on gas production from soluble fraction (GPSF) and gas production from non-soluble fraction (GPNSF).

### 3.3. Ruminal Fermentation Characteristics and CH_4_ Production

After 24 h of incubation time, the apparent dry matter degradability increased linearly (*p* ≤ 0.05) with the increasing ratio of artichoke bracts in the mixture (Table 6). The highest values (*p* < 0.05) of truly dry matter and organic matter degradability recorded with 500 g/kg of artichoke bracts in the mixture were 0.7045 and 0.6729 (g/g DM), respectively. Additionally, NDF degradability increased (L and Q; *p* ≤ 0.05) in response to increased artichoke bracts in the mixture. On the other hand, the PF decreased linearly (*p* ≤ 0.05) with the increasing ratio of artichoke bracts in the mixture (Table 6). CH_4_ production was not affected by forages but numerically decreased with 750 g/kg of forages. The concentration of NH_3_-N significantly increased with increasing artichoke bract content (L; *p* = 0.03), and a higher concentration was observed at 500 g/kg of forages (Q; *p* = 0.061). In terms of microbial protein, there were no significant differences between forages.

The ruminal pH of forages was significantly different (*p* ≤ 0.05) and depressed linearly (*p* = 0.007) with the increasing ratio of artichoke bracts in the mixture (Table 7). There were no significant differences in total VFA concentrations between forages. The molar proportions of VFA did not significantly differ by forage except for isobutyrate and isovalerate, which were reduced linearly (*p* ≤ 0.05) with increasing artichoke bract ratios in the mixture.

## 4. Discussion

The principle of silage making is based on preserving green fodder under anaerobic conditions to help the lactic acid-producing bacteria to generate lactic acid, resulting in a decrease in the pH values of silage as an indicator of high-quality fermentation in silage [41,42]. The growth of lactic acid bacteria is controlled by many factors such as temperature, the presence of sugars, and a_w_ [43]. In the current study, the temperature values were aptitude for the lactic acid bacteria growth in line with the earlier studies of Wang et al. [44] and Okoye et al. [43]. Comparably, Kung Jr et al. [45] indicated that the temperature of well-packed forages should not increase to more than 5 to 8 °C above the ambient temperature at filling, which was attained in our findings. The increase in temperature after extended ensiling times may be due to the reduced heat dissipation resulting from the larger forage mass working as an insulator. 

The values of a_w_ are suitable for the growth of lactic acid bacteria, as reported by Whiter and Kung Jr [46], who found that the growth of *L. plantarum* declined when the value decreased from 0.987 to 0.949. Additionally, our findings are similar to previous studies conducted on corn silage [47,48,49] in which the levels of a_w_ in silage were between 0.90 and 0.99. The increasing a_w_ of ensiled forages with ensiling times is related to the DM content of forages [50].

Compared with berseem silage, artichoke bracts silage has lower pH values that remained constant at less than 4.0 until 120 days of ensiling, which was indicative of well-preserved silage [51]. Adequate amounts of water-soluble carbohydrates could explain such results in artichoke bracts that are readily fermented by lactic acid-producing bacteria, causing the decrease in pH values in silage [24]. Furthermore, high phytogenic substance content in artichoke byproducts, such as terpenoids and polyphenols (e.g., caffeoylquinic acids, flavonoids) [21,52,53], may improve silage antioxidative potential and increase the abundance of lactic acid bacteria [54]. In the current study, the values of lactic acid in artichoke bracts containing silage were higher than those reported in the earlier studies of Meneses et al. [24], Meneses et al. [25], and Monllor et al. [53] and the recommended values (20–40 g/kg DM) for grass silages [45]. Those discrepancies could be explained by the differing DM content of forages (higher values with low DM content to <30), the initial lactic acid bacteria population, and the presence of fermentable carbohydrates [45,53].

Our results indicated that the highest decline of pH values is consistent with lactic acid observed during the first 30 days of ensiling for all ensiled forages, which is similar to that reported by Monllor et al. [53], who reported that the pH value of artichoke byproducts declined during the first 7 days of ensiling with increased lactic acid concentration. It is possible that artichoke increased the growth of heterofermentative lactic acid bacteria (e.g., Weissella, Lactococci, Leuconostocs, Pediococcus, and Enterococci) at the initial period of ensiling and then declined with ensiling time progression [55]. The lack of growth of heterofermentative lactic acid bacteria may result from the decreased polyphenol content of artichoke bracts at the end of ensiling. In this context, Monllor et al. [53] recorded the highest concentration of total polyphenols for artichoke byproducts at 30 and 60 days of ensiling compared to 200 days. In addition, heterofermentative lactic acid bacteria are sensitive to low pH [56].

Acetic acid levels in ensiled forages were within the recommended range (10–30 g/kg DM) [45]. Acetic acid plays a very important role in improving stability when silage is exposed to air; this is due to its strong inhibition of undesirable microorganisms (e.g., yeast and mold) [57]. On the other hand, reduced acetic acid concentration after mixing artichoke bracts with berseem may be attributed to its inhibitory effect on some acetic acid bacteria, such as enterobacteria, as confirmed by Monllor et al. [53]. These bacteria compete with lactic acid on fermented sugars, producing acetic acid, not lactic acid [57]. The value of recommended propionic acid in artichoke bracts silage was <1 g/kg DM for good silage. The high concentration of propionic acid of berseem silage (>5 g/kg DM) at 60 and 120 days could be explained by the growth of clostridia bacteria such as *Clostridium propionicum* [45]. The concentration of butyric acid was very low during 30 and 60 days of ensiling for high proportions of artichoke bracts in a mixture. This is similar to the finding of Meneses et al. [24]. Artichoke bracts showed strong inhibition of clostridia activity [53], rapidly reducing pH values throughout ensiling times [51]. The proteolysis processes during ensilage were inhibited by mixing artichoke bracts with berseem, as evidenced by the lower concentration of NH_3_-N, below 7 g kg^−1^ of total nitrogen at 30 and 60 days of ensiling. These are in harmony with the results reported by Meneses et al. [24] and are indicative of good-quality silage [58]. These results could be explained by the inhibition of proteolytic activity by clostridia [57], as protease enzyme activity was reduced at low pH 4.0 [59].

In this study, the initial DM content for fresh forages was similar to 250 g/kg, which was reported as the minimum DM content for proper silage [51]. The decline in DM content of ensiled forages over time could be due to the fermentation of silage DM by microflora that degrades nutrients into liquids, gases, and VFA (Desta et al., 2016. Moreover, the increase in DM content with increasing artichoke bracts ratios in the mixture could be explained by higher polyphenolic content that has an antimicrobial effect that could decrease the digestibility of nutrients (e.g., carbohydrates and proteins) in plants [15]. Further, the decline in pH in response to increased lactic acid concentrations with increasing artichoke ratio is considered the main factor for preserving the silage from undesirable bacterial growth that reduces the DM content of the silage. The differences in CP content of ensiled forages could be related to variations in proteolysis of various protein make-ups of ensiled forages due to the associative effect between artichoke and berseem that may affect the rate and extent of fermentation [51,60].

The increasing carbohydrate content with increasing ratios of artichoke bracts throughout the ensiling time is in harmony with the findings of Meneses et al. [24]. It could be explained by losses of water-soluble carbohydrates in the first phase of ensiling [61]. Moreover, the decrease in hemicellulose content of ensiled forages with the progression of ensiling times compared to day (0) is attributed to increased hemicellulose solubility because of the accumulation of organic acids during ensiling of forages, resulting in reduced hemicellulose content [62]. It can be concluded that 30 days of ensiling allowed for proper fermentation of all ensiled forages with the production of higher lactic acid and lower butyric acid and NH_3_-N concentrations, which agrees with the findings of Monllor et al. [53].

The higher cumulative gas production of artichoke silage compared to berseem silage may be attributed to their different fiber fraction [63]. Artichoke bracts have a lower lignin content (102 vs. 165 g/kg NDF) than berseem, which was found to have an antimicrobial effect [64]. Moreover, artichoke bracts have a higher content of rapidly fermented substrates created by rumen bacteria such as hemicellulose and water-soluble polysaccharide inulin (data not estimated) [65]. Additionally, artichoke bracts positively affected ruminal microbiota, as shown by the improvement in DM, OM, and NDF digestibility, which agrees with previous studies [25,53,66]. The digestibility of NDF in artichoke bracts was low DM (0.4171 g/gDM) compared to the value (0.592 g/gDM) reported by [66]. Variations in the cell wall content and nature of the NDF fraction of substrates appeared to cause the lack of data conformity in different studies [64].

The enhancement of fiber digestion by artichoke bracts in our study may be ascribed to the high NH_3_-N concentration used as a nitrogen source to grow cellulolytic bacteria [67]. The cellulolytic bacteria concentration in this study was within the recommended range (50–250 mg N/L) for maximum microbial growth and the maximum rate and extent of fermentation [68]. Moreover, the improvement of fiber digestibility may be due to pH, which was higher than 6.0, which is optimal for the growth of rumen cellulolytic bacteria [69]. The enhanced DM, OM, and fiber digestibility caused by the co-ensiling artichoke bracts with berseem can improve the efficiency of energy utilization and, consequently, animal performance.

The differences in rumen fermentation of various forage mixtures (berseem and artichoke) could be attributed to the associative effect between the forages. The forage mixture’s varying nature and chemical constituents cause asynchrony in nutrient releases, resulting in differences in microbial biomass growth [70]. 

Partitioning of the degraded substrate into gases (mainly CO_2_ and CH_4_), microbial mass, and short-chain fatty acids production was estimated by the PF. The PF is expressed as a ratio of organic matter degraded in vitro (mg) to the volume of the produced gas (mL) [71] and is used as an index of the efficiency of microbial mass synthesis (EMS) in vitro and to predict feed intake [39]. The value of PF obtained in the current study was within the theoretical range in roughages, which was from 2.75 to 4.45 mg/mL, which reflects YATP from 10 to 40 [39]. The inclusion of artichokes bracts into mixtures tended to reduce PF linearly. This means that partitioning of degraded matter to microbial mass decreased, while partitioning into CH_4_ production increased because of the high content of artichokes bracts of fermentable carbohydrates, which produce larger amounts of gas. This is in accordance with Rymer and Givens [72], who reported that roughages with high fermentable carbohydrates (e.g., sugarbeet feed, grass silage, molasses, wheat, and maize) release larger gas amounts and VFA, which yield smaller microbial mass amounts. The relationship between biomass yield and gas produced per unit of truly degraded substrate was inversely related (r = −0.78) [70]. 

In the current study, CH_4_ emission was not affected by forage mixtures except for the 750 g/kg forage that decreased CH_4_ emission numerically. This could be explained by specific factors (e.g., type of forage, chemical composition, and maturity) and the quality of the fermentation process during silage making [73,74]. The increase in the enhancement of fiber digestion by artichoke bracts in our study may be ascribed to the high NH_3_-N concentration used as a nitrogen source. To grow cellulolytic bacteria concentration by increasing the artichoke bracts ratio indicates a higher content of artichoke with rumen degradable protein than berseem, although berseem contains higher CP.

## 5. Conclusions

The current study findings concluded that mixing artichoke bracts with berseem, particularly at intermediate ratios of artichoke bracts, enhanced silage fermentation quality, and the optimum fermentation time was found to be 30 days of ensiling. Additionally, an artichoke bracts concentration of 500 g/kg of fresh forage improved rumen degradation in vitro. Our findings provide a theoretical basis for producing high-quality silage from legume forages such as berseem when mixed with artichoke bracts at a ratio of 1:1 to minimize nutrient wastage and increase livestock sustainability. Further in vivo investigations are required to evaluate the palatability of artichoke bracts silage by animals and its effect on animal performance.

## Figures and Tables

**Figure 1 animals-13-01543-f001:**
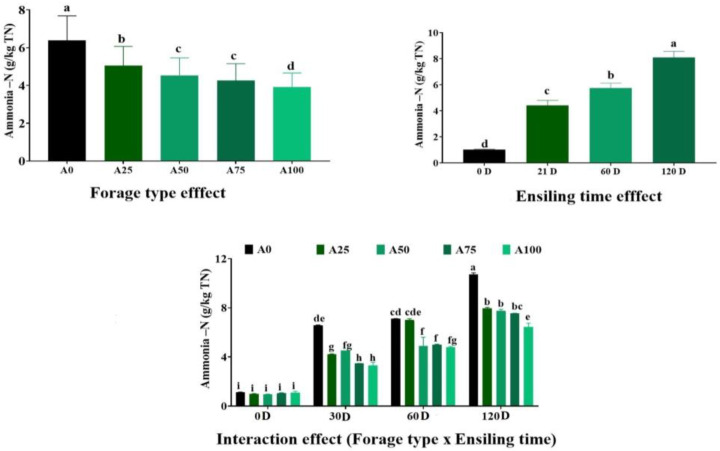
Ammonia–N (g/kg TN) of berseem ensiled with artichoke bracts at different levels after 0, 30, 60 and 120 days of ensiling. ^a–g^ Means in a bar with different superscripts differ significantly. Berseem and artichoke bracts were mixed before ensiling at ratios 100:0 (A0), 75:25 (A25), 50:50(A50), 25:75 (A75) and 0:100 (A100, wt:wt on fresh weight basis). The values shown are the means ± SE.

**Table 1 animals-13-01543-t001:** Physical properties of berseem ensiled with artichoke bracts at different levels after 0, 30, 60 and 120 days of ensiling.

Item	Days of Ensiling ^1^	Forage Type ^2^					SEM ^3^	*p*-Value ^4^		
		A0	A25	A50	A75	A100		L	Q	C
Temperature (°C)										
	0	22.5	22.5	22.4	22.5	22.3	0.10	0.44	0.21	0.60
	30	25.6	26.2	25.6	26	25.8		0.53	0.71	0.50
	60	29.4	29.6	29.6	29.5	29.5		0.60	0.64	0.94
	120	29.8	29.4	29.7	29.4	29.6		0.30	0.80	0.40
Water activity (aw)										
	0	0.977	0.973	0.968	0.964	0.959	0.04	0.13	0.46	0.02
	30	0.955	0.967	0.989	0.974	0.973		0.11	0.32	0.62
	60	0.973	0.979	0.975	0.983	0.972		0.30	0.80	0.80
	120	0.978	0.986	0.972	0.979	0.984		0.50	0.50	0.20
pH										
	0	6.0 ^aA^	5.9 ^abA^	5.9b ^cA^	5.8 ^cA^	5.4 ^dA^	0.04	0.64	0.01	<0.01
	30	4.5 ^aC^	4.0 ^bB^	3.9 ^bB^	4.0 ^bB^	4.1 ^bB^		<0.01	0.02	0.53
	60	4.9 ^aB^	4.3 ^bB^	4.0 ^cB^	3.8 ^cB^	3.8 ^cC^		<0.01	0.03	<0.01
	120	4.9 ^aB^	4.3 ^bB^	4.3 ^bB^	3.9 ^bB^	3.8 ^bC^		0.02	0.20	<0.01

^A–C^ Means in the same column followed by different superscripts differ (*p* < 0.05). ^a–d^ Means in the same row followed by different superscripts differ (*p* < 0.05). ^1^ Time of ensiling 0, 30, 60 and 120 days. ^2^ Berseem and artichoke bracts were mixed before ensiling at ratios 100:0 (A0), 75:25 (A25), 50:50 (A50), 25:75 (A75) and 0:100 (A100, wt:wt on fresh weight basis). ^3^ SEM—standard error of means. ^4^
*p*-values are shown for L—linear, and Q—quadratic. C—cubic effects.

**Table 2 animals-13-01543-t002:** Organic acids and ethanol (g/kg DM) of berseem ensiled with artichoke bracts at different levels after 0, 30, 60 and 120 days of ensiling.

Item ^1^	Days of Ensiling ^1^	Forage Type ^2^					SEM ^3^	*p*-Value ^4^		
		A0	A25	A50	A75	A100		L	Q	C
Lactic acid										
	0	8.8 ^bD^	8.4 ^bC^	6.0 ^cB^	11.1 ^aD^	12.0 ^aD^	0.13	0.49	0.06	<0.01
	30	57.5 ^cA^	131.6 ^aA^	133.5 ^aA^	125.2 ^aB^	107.7 ^bB^		<0.01	<0.01	0.27
	60	19.1 ^bC^	18.0 ^bBC^	119.4 ^aA^	128.5 ^aB^	126.0 ^aA^		<0.01	0.20	<0.01
	120	25.6 ^bB^	20.9 ^cB^	17.4 ^dB^	67.2 ^aC^	68.4 ^aC^		<0.01	<0.01	<0.01
Acetic acid										
	0	0 ^aC^	0 ^aB^	0 ^aB^	0 ^aC^	0 ^aB^	0.1	-	-	-
	30	26.6 ^aA^	26.7 ^aA^	9.8 ^bA^	8.5 ^bB^	5.9 ^bAB^		0.02	0.90	0.05
	60	19.4 ^aB^	19.3 ^aA^	13.8 ^bA^	11.4 ^bAB^	9.8 ^bA^		0.04	0.90	0.01
	120	22.9 ^aAB^	22.7 ^aA^	15.8 ^abA^	14.9 ^abA^	7.7 ^bAB^		0.30	0.50	002
Propionic acid										
	0	0.00 ^aC^	0.00 ^aC^	0.00 ^aC^	0.00 ^aC^	0.00 ^aC^	0.02	-	-	-
	30	0.29 ^aC^	0.00 ^bC^	0.09 ^bB^	0.004 ^bB^	0.001 ^bB^		-	-	-
	60	1.9 ^aB^	2.19 ^aB^	0.092 ^bB^	0.002 ^bB^	0.04 ^bB^		-	-	-
	120	7.43 ^aA^	3.02 ^bA^	0.4 ^cA^	0.04 ^cA^	0.11 ^cA^		<0.01	<0.01	<0.01
Butyric acid										
	0	0.00 ^C^	0.00 ^C^	0.00 ^D^	0.00 ^B^	0.00 ^C^	0.08	-	-	-
	30	4.7 ^aC^	0.00 ^cC^	1.3 ^bC^	0.00 ^cB^	0.00 ^cC^		<0.01	<0.01	<0.01
	60	28.8 ^aA^	25.0 ^aA^	4.6 ^bB^	0.00 ^bB^	1. 0 ^bD^		<0.01	<0.01	<0.01
	120	17.5 ^aB^	13.1 ^aB^	14.4 ^aA^	4.3 ^bA^	4.00 ^bA^		0.06	0.90	<0.01
Ethanol										
	0	0.00 ^aC^	0.00 ^aC^	0.00 ^aB^	0.00 ^aC^	0.00 ^aB^	0.09	-	-	-
	30	18.6 ^aAB^	19.4 ^aA^	15.9 ^aA^	12.8 ^aB^	14.3 ^aA^		0.24	0.51	0.28
	60	25.7 ^aA^	17.7 ^abAB^	21.0 ^abA^	20.0 ^abA^	15.8 ^bA^		0.40	0.70	0.70
	120	11.2 ^cB^	14.8 ^abcB^	16.5 ^abA^	18.6 ^aA^	12.4 ^bcA^		<0.01	0.80	0.80

^A–D^ Means in the same column followed by different superscripts differ significantly. ^a–c^ Means in the same row followed by different superscripts differ significantly. ^1^ Time of ensiling: 0, 30, 60 and 120 days. ^2^ Berseem and artichoke bracts were mixed before ensiling at ratios 100:0 (A0), 75:25 (A25), 50:50 (A50), 25:75 (A75), and 0:100 (A100, wt:wt on fresh weight basis).^3^ SEM—standard error of means. ^4^
*p*-values are shown for L—linear and Q—quadratic. C—cubic effects.

**Table 3 animals-13-01543-t003:** Chemical composition (g/kg DM) of berseem ensiled with artichoke bracts at different levels after 0, 30, 60, and 120 days of ensiling.

Item ^1^	Days of Ensiling ^1^	Forage Type ^2^					SEM ^3^	*p*-Value ^4^		
		A0	A25	A50	A75	A100		L	Q	C
Dry matter										
	0	252.9 ^aA^	252.4 ^aA^	251.9 ^aA^	251.3 ^aA^	250.8 ^aA^	0.14	0.87	0.93	0.76
	30	168.0 ^cB^	179.6 ^cB^	201.1 ^bB^	217.4 ^aB^	225.3 ^aB^		<0.01	0.14	<0.01
	60	176.8 ^cB^	174.2 ^cB^	190.1 ^bC^	194.9 ^bC^	208.4 ^aB^		0.03	0.65	<0.01
	120	152.8 ^cC^	174.6 ^cB^	178.8 ^bcD^	193.9 ^abC^	202.8 ^aB^		<0.01	0.02	<0.01
Organic matter										
	0	884.7 ^cA^	898.5 ^bcA^	912.4 ^abcA^	926.2 ^abA^	940.0 ^aA^	0.18	0.03	0.26	<0.01
	30	880.3 ^dA^	895.5 ^cdAB^	904.4 ^bcA^	917.7 ^abA^	928.3 ^aA^		<0.01	0.05	<0.01
	60	882.5 ^cA^	887.1 ^cC^	907.1 ^bA^	922.4 ^aA^	935.3 ^aA^		<0.01	0.47	<0.01
	120	855.4 ^eB^	890.1 ^dBC^	902.2 ^cA^	922.7 ^bA^	938.2 ^aA^		<0.01	<0.01	<0.01
Crude protein										
	0	164.5 ^aAB^	156.5 ^bA^	148.5 ^cA^	140.5 ^dC^	132.5 ^eA^	0.11	<0.01	<0.01	<0.01
	30	176.1 ^aA^	155.6 ^bA^	138.8 ^cB^	143.6 ^bcB^	141.4 ^bcA^		<0.01	<0.01	0.02
	60	158.9 ^aAB^	147.1 ^abA^	151.2 ^abA^	140.2 ^bC^	140.1 ^bA^		0.09	0.22	<0.01
	120	145.5 ^abB^	143.6 ^abA^	134.9 ^bB^	150.4 ^aA^	139.0 ^abA^		0.49	0.02	0.15
Neutral detergent fiber										
	0	519.6 ^eB^	539.8 ^dC^	560.1 ^cB^	580.3 ^bB^	600.5 ^aC^	0.16	<0.01	<0.01	<0.01
	30	493.9 ^dC^	565.4 ^cB^	568.4 ^cB^	609.6 ^bA^	639.6 ^aA^		<0.01	<0.01	<0.01
	60	535.0 ^cAB^	610.4 ^bA^	564.2 ^bB^	579.1 ^aB^	622.9 ^aB^		0.04	<0.01	<0.01
	120	539.3 ^bA^	555.0 ^bBC^	589.8 ^aA^	604.8 ^aA^	598.4 ^aC^		<0.01	0.48	<0.01
Acid detergent fiber										
	0	294.1 ^eD^	308.7 ^dC^	323.2 ^cC^	337.8 ^bC^	352.3 ^aC^	0.12	<0.01	<0.01	<0.01
	30	330.8 ^dC^	367.8 ^cB^	361.0 ^cB^	388.9 ^bA^	408.5 ^aA^		<0.01	<0.01	<0.01
	60	366.0 ^bB^	401.5 ^aA^	368.7 ^bB^	367.7 ^bB^	390.1 ^aB^		0.14	<0.01	<0.01
	120	384.8 ^aA^	267.8 ^bD^	388.2 ^aA^	381.8 ^aAB^	390.1 ^aB^		0.30	<0.01	<0.01
Acid detergent lignin										
	0	58.4 ^aC^	56.3 ^abC^	54.2 ^bcD^	52.1 ^bcA^	56.0 ^cA^	0.12	0.04	0.30	<0.01
	30	81.6 ^aB^	75.1 ^abAB^	64.7 ^abB^	59.3 ^bA^	65.3 ^abA^		<0.01	0.89	0.09
	60	93.6 ^aA^	89.3 ^aA^	61.1 ^bC^	57.5 ^bA^	56.1 ^bA^		<0.01	0.58	<0.01
	120	91.6 ^aA^	70 ^bB^	70.2 ^bA^	56.3 ^cA^	62.9 ^bcA^		<0.01	0.04	<0.01
Cellulose										
	0	235.7 ^cD^	252.4 ^dC^	269.0 ^cB^	285.7 ^bA^	302.3 ^aC^	0.14	<0.01	<0.01	<0.01
	30	249.2 ^cC^	292.6 ^bB^	296.3 ^bA^	329.6 ^aB^	343.2 ^aA^		<0.01	<0.01	<0.01
	60	272.5 ^cB^	312.2 ^bA^	307.5 ^bA^	310.2 ^bC^	334.0 ^aAB^		<0.01	<0.01	<0.01
	120	293.3 ^bA^	197.8 ^cD^	316.0 ^abA^	325.5 ^aB^	327.2 ^aB^		<0.01	<0.01	<0.01
Hemicelluloses										
	0	225.5 ^eA^	231.2 ^dB^	236.9 ^cA^	242.5 ^bC^	248.2 ^aA^	0.12	<0.01	<0.01	<0.01
	30	163.1 ^dB^	197.7 ^cC^	207.4 ^bcB^	220.7 ^abA^	231.1 ^aB^		<0.01	<0.01	<0.01
	60	168.9 ^dB^	208.9 ^bcC^	195.5 ^cB^	211.5 ^bB^	232.8 ^aB^		<0.01	<0.01	<0.01
	120	154.3 ^dC^	287.2 ^aA^	203.7 ^cB^	223.0 ^bAB^	208.2 ^bcC^		<0.01	<0.01	<0.01

^A–D^ Means in the same column followed by different superscripts differ (*p* < 0.05). ^a–e^ Means in the same row followed by different superscripts differ (*p* < 0.05). ^1^ Time of ensiling 0, 30, 60 and 120 days. ^2^ Berseem and artichoke bracts were mixed before ensiling at ratios 100:0 (A0), 75:25 (A25), 50:50 (A50), 25:75 (A75) and 0:100 (A100, wt:wt on fresh weight basis). ^3^ SEM—standard error of means. ^4^
*p*-values are shown for L—linear and Q—quadratic. C—cubic effects.

**Table 4 animals-13-01543-t004:** Cumulative gas production (mL/g DM) of berseem ensiled with artichoke bracts at different levels after 30 days of ensiling incubated buffered rumen fluid in vitro for 72 h.

Incubation Time (h)	Forage Type ^1^				
	A0	A25	A50	A75	A100
3	26.25	25.29	25.00	24.44	22.32
6	50.32	44.34	62.23	45.04	50.69
9	70.70	66.67	89.07	63.35	68.94
12	93.63	86.77	107.36	83.29	92.95
24	143.06	133.07	163.32	152.03	163.63
48	174.94	163.19	207.99	192.77	201.63
72	187.06	176.61	226.51	214.15	222.41
Overall mean	92.40^c^	99.42 ^bc^	114.14 ^a^	98.55 ^bc^	101.76 ^b^
SEM	6.06	9.25	7.17	7.16	8.01

^a–c^ Means in the same row carry different superscripts that are significantly different (*p* < 0.05) for forage type effect. *p*-values for the effect of forage, time, and forage × time interaction was <0.01, <0.01, and 0.2609, respectively. *p*-values are shown for L, linear (0.078) and Q, quadratic (0.4867) and cubic (0.1968) effects. SEM—standard error of means. ^1^ Berseem and artichoke bracts were mixed before ensiling at ratios 100:0 (A0), 75:25 (A25), 50:50(A50), 25:75 (A75), and 0:100 (A100, wt:wt on fresh weight basis).

**Table 5 animals-13-01543-t005:** Parameters of gas production (mL/200 mgDM) of berseem ensiled with artichoke bracts at different levels after 30 days of ensiling incubated buffered rumen fluid in vitro for 72 h.

Forage ^1^	Parameters of Gas Production ^2^				
	a (mL)	b (mL)	c (mL/h)	GPSF	GPNSF
A0	−0.379	39.43	0.057	24.18	127.25
A25	−0.394	36.32	0.054	22.04	111.93
A50	−0.634	47.52	0.072	26.59	157.89
A75	−0.318	42.30	0.041	20.90	114.53
A100	−0.585	48.71	0.056	23.82	154.98
S.E.M	−0.207	1.78	0.003	0.77	7.69
*p*-value	0.99	0.06	0.06	0.16	0.12
Trend analysis					
Linear	0.84	0.03	0.42	0.69	0.22
Quadratic	0.98	0.81	0.52	0.98	0.68
Cubic	0.83	0.76	0.15	0.67	0.60

^1^ Berseem and artichoke bracts were mixed before ensiling at ratios 100:0 (A0), 75:25 (A25), 50:50 (A50), 25:75 (A75) and 0:100 (A100, wt:wt on fresh weight basis); SEM—standard error of means. ^2^ a—gas production from the soluble fraction (mL); b—gas production from the insoluble fraction (mL); c—gas production rate (mL/h); GPSF—gas production from soluble fraction; GPNSF—gas production from non-soluble fraction.

**Table 6 animals-13-01543-t006:** Effect of berseem ensiled with artichoke bracts at different levels after 30 days of ensiling on feed degradability, partitioning factor (PF), methane production (CH_4_), ammonia–nitrogen (NH_3_-N) and microbial protein (MP) at 24 h of incubation time.

Forage ^1^	Feed Degradability (g/g DM) ^2^				PF (mg/mL Gas)	CH_4_ (mL/g DM Incubated)	NH_3_-N (mg/100 mL)	MP (g/kg DOM)
	Dry Matter		TDOM	NDFD				
	Apparent	True						
A0	0.5596	0.6571 ^b^	0.6223 ^b^	03369	3.44	50.75	23.52 ^b^	118.24
A25	0.5416	0.6347 ^b^	0.5983 ^b^	0.3714	3.64	46.86	23.89 ^b^	113.68
A50	0.6384	0.7045 ^a^	0.6729 ^a^	0.4849	3.41	63.01	27.00 ^a^	127.85
A75	0.5717	0.6559 ^b^	0.6305 ^ab^	0.4486	3.45	36.74	25.01 ^ab^	119.80
A100	0.6253	0.6641 ^b^	0.6361 ^ab^	0.4171	3.06	69.47	25.36 ^ab^	113.09
S.E.M	1.36	0.79	0.85	2.03	0.07	4.33	0.39	2.08
*p*-value	0.06	0.046	0.050	0.066	0.091	0.177	0.017	0.185
Trend analysis								
Linear	0.039	0.350	0.167	0.046	0.031	0.314	0.037	0.735
Quadratic	0.792	0.221	0.255	0.049	0.071	0.351	0.061	0.113
Cubic	0.936	0.371	0.255	0.486	0.996	0.199	0.860	0.213

^a,b^ Means in the same column carry different superscripts that are significantly different (*p* < 0.05). ^1^ Berseem and artichoke bracts were mixed before ensiling at ratios 100:0 (A0), 75:25 (A25), 50:50 (A50), 25:75 (A75), and 0:100 (A100, wt:wt on fresh weight basis) after 30 days of ensiling; SEM—standard error of means. ^2^ TDOM—truly degraded organic matter; NDFD—neutral detergent fiber degraded.

**Table 7 animals-13-01543-t007:** Ruminal pH and total and molar proportions of volatile fatty acids (VFA) profiles of berseem ensiled with artichoke bracts at different levels after 30 days of ensilage after incubation for 24 h.

Forage ^1^	pH	VFA Molar Proportions ^2^						C2/C3 ^3^	Total VFA (mM)
		C2	C3	C4	C4i	C5i	C5		
A0	6.88 ^a^	63.10	17.19	11.80	3.07	2.59	1.32	3.67	99.07
A25	6.66 ^b^	63.95	17.82	12.20	3.34	2.59	1.45	3.60	101.36
A50	6.69 ^b^	66.81	18.04	12.24	3.12	2.53	1.35	3.71	104.08
A75	6.63 ^b^	65.03	17.32	11.96	2.89	2.32	1.27	3.76	100.79
A100	6.64 ^b^	62.04	16.26	11.40	2.69	2.22	1.24	3.81	95.85
S.E.M	0.04	0.86	0.33	0.18	0.09	0.16	0.05	0.04	1.48
*p*-value	0.01	0.59	0.65	0.72	0.244	0.118	0.520	0.677	0.59
Trend analysis									
Linear	0.007	0.873	0.366	0.462	0.055	0.0143	0.207	0.196	0.429
Quadratic	0.060	0.150	0.194	0.222	0.187	0.390	0.330	0.591	0.152
Cubic	0.288	0.651	0.977	0.949	0.398	0.696	0.356	0.606	0.896

^a,b^ Means in the same column carry different superscripts that are significantly different (*p* < 0.05). ^1^ Berseem and artichoke bracts were mixed before ensiling at ratios 100:0 (A0), 75:25 (A25), 50:50 (A50), 25:75 (A75) and 0:100 (A100, wt:wt on fresh weight basis) after 30 days of ensiling; SEM, standard error of means. ^2^ C2, acetate; C3, Propionate; C4, butyrate; C4i, isobutyrate; C5i isovalerate; C5, valerate. ^3^ C2/C3 ratio acetate-to-propionate ratio.

## Data Availability

The datasets used along with this research are available from the corresponding author upon reasonable request.

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
