# Peer review of "Ensiling Characteristics, In Vitro Rumen Fermentation Patterns, Feed Degradability, and Methane and Ammonia Production of Berseem (Trifolium alexandrinum L.) Co-Ensiled with Artichoke Bracts (Cynara cardunculus L.)"

_animals, 2023, doi:10.3390/ani13091543_

Round 1

Reviewer 1 Report

This study evaluated the effect of co-ensiling berseem (Trifolium alexandrinum L.) with artichoke bracts (Cynara cardunculus L.) on silage characteristics, in vitro fermentation, degradability and methane and ammonia production. The authors found that increasing the level of artichoke in the silage led to an improvement in silage quality, as portrayed in chemical composition and digestibility, with the best quality obtained in the 50:50 forage mixture. This study provides useful information that will contribute to the efficient utilization of plant byproducts as ruminant feed. The authors conducted an in-depth evaluation, from fresh forage quality to ensiling characteristics and the final product in vitro digestibility. The manuscript is well-written and interesting to read, and the authors have made an appreciable effort to use the most recent references.

However, the quality of the manuscript can be improved by addressing the following suggestions.

Abstract

Please highlight the practical implication of this study to the intended audience in the abstract.

PF in Line 34 should be defined or written in full for clarity.                                

Introduction

Please define abbreviations when they first appear in the main body—for instance, DM in Line 59.

The authors have not clearly outlined the research question they intended to address in this study. Are there previous studies that have examined the effect of co-ensiling artichoke with a legume? If no, please indicate this as a research gap that this study aims to fill.

The reason behind the choice of berseem (Trifolium alexandrinum L.) and not any other legume available in the region should be highlighted.

Materials and methods

Please provide the common name of Trifolium alexandrinum where it first appears in Lines 78-79) and just use the common name henceforth.

The scientific name of Cynara cardunculus L. is already provided in Line 61. Using just the common name in Line 90 will be sufficient.

Highlighting some details on how the forages were produced will be useful. For instance, were they under irrigation or just rain-fed, were they fertilized, was the berseem grown commercially or in experimental plots, etc.

The definition of DM should be in Line 59, where it first appeared, and not in Line 125.

“Before the day experiment” appears twice in the sentence (Lines 144-145).

The treatment lables, e.g. A0, A25…. etc., can be used here without the ratios because they are already defined in Lines 94 and 95.

Please ensure that the equation in Line 173 is presented correctly, as it appears in the source cited. Using the same letter (t) in the same equation to represent different components is confusing to the reader.

The authors refer to section 2.1.2 in Line 193, but there is no such section in the manuscript. Please double-check.

Results

Please avoid describing results in the Table footnotes. E.g. Lines 228-231, “Effects of forage, ensiling time and forage × time interaction were significant (p<0.01) for pH values. Effects of forage and forage × time interaction were not significant (P > 0.05) for temperature and water activity. Effect of ensiling time was significant (P<0.05) for temperature and water activity”, and Line 276-278 “All effects of forage, ensiling time and forage × time interaction were significant (p<0.01) except for ethanol the effect of forage was not significant (P=0.13)”. Kindly check the other tables too.

It is not clear what message the authors were presenting in Lines 243-245, “Significant effects of forage type, ensiling time, and their interaction on the concentration of VFA (P<0.05)”. The sentence may be incomplete.

Please be consistent with use of abbreviations or full words but not both. For instance, ammonia-N is used in both forms in Lines 255-257.

Please ensure that interpretation of the Tables is accurate. For instance, Lines 264-266 reads “The content of DM and OM linearly increased (L, P<0.01), while CP decreased with an increasing proportion of artichoke bracts in the forage mixtures.” However, Table 3 shows no linear increase in DM with an increase in the proportion of artichoke bracts for the day 0 forage. Also, the quadratic response described for day 60 in Lines 267-268 for CP is not significant in Table 3.

The acetic acid, propionic acid and ethanol are all presented as 0 on day 0 for all forage mixes (Table 2). Do these zeros mean that the variables were not detected? If yes, then it is unclear why there are superscripts in the rows, considering we can not conduct statistical analysis on all zero figures. If no, then it is understandable that the values were negligible (when rounded off to two decimal points they still appeared as 0) and not statistically different.

Although Figure 1 panels look complete, their interpretation will be easier if the x-axis is labelled. Alternatively, the centre justification of panel C should occur in the panel title (Interaction effect (Forage type x Ensiling time) as well for clarity.

Table 4: Please double-check the table title. Also, it is unclear what the “Main effect” row represents and if the superscript letters show differences between forage type means. Is it possible to present the statistical output clearly as in Table 5?

Table 6: The table title indicates “30 days of ensiling”, while the footnote indicates “21 days of ensiling”. Please check Table 7 too.

Discussion

Line 378: Please do not refer to tables in the discussion.

Please do not define abbreviations presented in the main text already, and avoid using full words that have been abbreviated already: E.g. Line 444 (ammonia), 467 (partitioning factor (PF)).

Please highlight what the main findings mean regarding feeding value to the target animals. For instance,  what would the improved silage digestibility mean when feeding an animal? Does it mean better utilization of the feed…?

Conclusions

Please highlight the practical implications of this study.

Author Response

Dear Reviewer 1#

Thank you for your comments concerning our manuscript, These comments are valuable and very helpful for revising and improving our paper, as well as the important guiding significance to our research. We have studied the comments carefully and have made the revisions that are highlighted in red in the manuscript, and we hope these revisions can meet with approval. Our responses are as flowing:

Abstract

Point 1: Please highlight the practical implication of this study to the intended audience in the abstract.

Response 1: We are grateful to this very valuable suggestion. We have added the practical implication in the abstract (Lines 35-38) to be as follows “In summary, co-ensiling artichoke bracts with berseem at a ratio of 1:1 might be a promising and easy method for the production of high-quality silage from legume forage with positively manipulating rumen fermentation”.

Point 2: PF in Line 34 should be defined or written in full for clarity.  

Response 2: The full term has been added as directed (Line 34).

Introduction

Point 3: Please define abbreviations when they first appear in the main body—for instance, DM in Line 59.

Response 3: Thanks for the reviewer’s valuable comment and suggestion. All abbreviations have been defined when they first appear throughout the manuscript (Line 65).

Point 4: The authors have not clearly outlined the research question they intended to address in this study. Are there previous studies that have examined the effect of co-ensiling artichoke with a legume? If no, please indicate this as a research gap that this study aims to fill.

Response 4: Thank you for draw our attention to clarify this important point. This point has been clarifed at the end of the introduction (Lines 83-84).

Point 5: The reason behind the choice of berseem (Trifolium alexandrinum L.) and not any other legume available in the region should be highlighted.

Response 5: We greatly appreciate this constructive comment.The reason for choosing berseem (Trifolium alexandrinum L.) has been clarified in the introduction section (Lines 45-49).

Point 6: Materials and methods; Please provide the common name of Trifolium alexandrinum where it first appears in Lines 78-79) and just use the common name henceforth.

Response 6: Thanks a lot for this valuable comment. The scientific name of Trifolium alexandrinum has been mentioned it first appears (in Lines 45) and hereafter we have used the common name according to your recommendations.

Point 7: The scientific name of Cynara cardunculus L. is already provided in Line 61. Using just the common name in Line 90 will be sufficient.

Response 7: Done as directed (Line 67).

Point 8: Highlighting some details on how the forages were produced will be useful. For instance, were they under irrigation or just rain-fed, were they fertilized, was the berseem grown commercially or in experimental plots, etc.

Response 8: Thanks a lot for this important comment. The required details have been added in lines 96-97.

Point 9: The definition of DM should be in Line 59, where it first appeared, and not in Line 125.

Response 9: Done as directed (Lines 65 and 132).

Point 10: “Before the day experiment” appears twice in the sentence (Lines 144-145).

Response 10: Corrected as directed (Lines 151-152).

 Point 11: The treatment lables, e.g. A0, A25…. etc., can be used here without the ratios because they are already defined in Lines 94 and 95.

Response 11: Done as directed (Lines 161-162).

Point 12: Please ensure that the equation in Line 173 is presented correctly, as it appears in the source cited. Using the same letter (t) in the same equation to represent different components is confusing to the reader.

Response 12: Thanks a lot for this important comment. The equation has been rewritten for more clarity to the journal readers (Lines 179-182).

Point 13: The authors refer to section 2.1.2 in Line 193, but there is no such section in the manuscript. Please double-check.

Response 13: It was a typing error and it has been corrected in Lines 199-200 to “as defined in section 2.2.”

Point 14: Results; Please avoid describing results in the Table footnotes. E.g. Lines 228-231, “Effects of forage, ensiling time and forage × time interaction were significant (p<0.01) for pH values. Effects of forage and forage × time interaction were not significant (P > 0.05) for temperature and water activity. Effect of ensiling time was significant (P<0.05) for temperature and water activity”, and Line 276-278 “All effects of forage, ensiling time and forage × time interaction were significant (p<0.01) except for ethanol the effect of forage was not significant (P=0.13)”. Kindly check the other tables too.

Response 14: Thanks lot for this constructive comment and valuable suggestion. All tables footnotes have been revised and all repetion of the results have been removed as directed.

Point 15: It is not clear what message the authors were presenting in Lines 243-245, “Significant effects of forage type, ensiling time, and their interaction on the concentration of VFA (P<0.05)”. The sentence may be incomplete.

Response 15: Thanks a lot for this valuable observation. The sentence has been completed (Lines 246-248).

Point 16: Please be consistent with use of abbreviations or full words but not both. For instance, ammonia-N is used in both forms in Lines 255-257.

Response 16: Corrected throughout the manuscript as directed (Line 259).

Point 17: Please ensure that interpretation of the Tables is accurate. For instance, Lines 264-266 reads “The content of DM and OM linearly increased (L, P<0.01), while CP decreased with an increasing proportion of artichoke bracts in the forage mixtures.” However, Table 3 shows no linear increase in DM with an increase in the proportion of artichoke bracts for the day 0 forage. Also, the quadratic response described for day 60 in Lines 267-268 for CP is not significant in Table 3.

Response 17: We are extremely grateful to the reviewer for pointing out this point. The interpretations of the tables have been revised and corrected as directed (Lines 267-276).

Point 18: The acetic acid, propionic acid and ethanol are all presented as 0 on day 0 for all forage mixes (Table 2). Do these zeros mean that the variables were not detected? If yes, then it is unclear why there are superscripts in the rows, considering we can not conduct statistical analysis on all zero figures. If no, then it is understandable that the values were negligible (when rounded off to two decimal points they still appeared as 0) and not statistically different.

Response 18: Thanks a lot for this comment. The acetic acid, propionic acid, and ethanol were analyzed and detected at the 0 day of ensiling. We added the statistical letters of significance as we analyzed the data for both effects of the ensilage time and forage type. Thus, there were significant differences when we compare their results for 0 days compared to the other time points.

Point 19: Although Figure 1 panels look complete, their interpretation will be easier if the x-axis is labelled. Alternatively, the centre justification of panel C should occur in the panel title (Interaction effect (Forage type x Ensiling time) as well for clarity.

Response 19: Thanks a lot for this constructive suggestion. We have added the lable at X axis in Fig. 1 as directed.

Point 20: Table 4: Please double-check the table title. Also, it is unclear what the “Main effect” row represents and if the superscript letters show differences between forage type means. Is it possible to present the statistical output clearly as in Table 5?

Response 20: Thank you for pointing this out.The title has been rewritten. The main effect has bee replaced with overall mean. The meaning of the supercript letters have been clarified as directed (Lines 308-311).

Point 21: Table 6: The table title indicates “30 days of ensiling”, while the footnote indicates “21 days of ensiling”. Please check Table 7 too.

Response 21: It was a typing error and it has been corrected to “after 30 days of ensiling” (Line 352).

Point 22: Discussion; Line 378: Please do not refer to tables in the discussion.

Response 22: Thank you. The mention of all tables has been removed from the discussion section.

Point 23: Please do not define abbreviations presented in the main text already, and avoid using full words that have been abbreviated already: E.g. Line 444 (ammonia), 467 (partitioning factor (PF)).

Response 23: Done throughout the manuscript as directed.

Point 24: Please highlight what the main findings mean regarding feeding value to the target animals. For instance, what would the improved silage digestibility mean when feeding an animal? Does it mean better utilization of the feed…?

Response 24: Thank you for pointing this out.This point has been clarified in lines 465-467.

Point 25: Conclusions; Please highlight the practical implications of this study.

Response 25: Thanks for the reviewer’s valuable comments and suggestions. We have updated the conclusion with the practical implications of the study in Lines 500-502 .

Reviewer 2 Report

1. Introduction

In this part, the authors stated the background of artichoke bracts (Cynara car-dunculus L.), however missed to provide the relevant background about berseem (Trifolium alexandrinum L) and relevant reference.

2. Materials and methods

Line 89: The growth stage or maturity of Berseem (Trifolium alexandrinum L.) harvested in this study should be stated.

Line 171-175. How to measure the soluble fraction and insoluble fraction?

Line 177-179: Why did the gas was collected only 1 mL from incubation and transferred into 5 mL glass vacuum airtight tubes? And ? mL gas was injected into the GC? And why didn`t collected the gas samples at 48 and 72 hours for methane detection?

3. Results

Table 4. How to calculate Main effect?

Line 320. GPSF and GPNSF should be clarified when first used.

Table 6: TOMD or TDOM?

Table 4, Table 5, Table 6:

In this study, cumulative gas production was measured at different incubation times, and parameters of gas production were measured at 72 h. And In table 6, PF, methane production, ammonia-nitrogen, and microbial protein were measured at 24 h of incubation time. Why were the times of measurements different?

Author Response

Dear Reviewer 2#

Thank you for your comments concerning our manuscript, these comments are valuable and very helpful for revising and improving our paper, as well as the important guiding significance to our research. We have studied the comments carefully and have made the revisions that are highlighted in red in the manuscript, and we hope these revisions can meet with approval. Our responses are as flowing:

Point 1: Introduction: In this part, the authors stated the background of artichoke bracts (Cynara car-dunculus L.), however missed to provide the relevant background about berseem (Trifolium alexandrinum L) and relevant reference.

Response 1: Thank you for carefully reviewing our manuscript and for this constructive suggestion. The background behind choosing berseem (Trifolium alexandrinum L.) has been clarified in the introduction section (Lines 45-49).

Point 2: Materials and methods; Line 89: The growth stage or maturity of Berseem (Trifolium alexandrinum L.) harvested in this study should be stated.

Response 2: Thanks a lot for this important comment. The growth stage of berseem (Trifolium alexandrinum L.) collected samples was the 4th cut. More details about berseem samples have been added in lines 96-97.

Point 3: Line 171-175. How to measure the soluble fraction and insoluble fraction?

Response 3: Thanks a lot for this valuable comment. All gas production kinetics including the gas produced from the soluble or insoluble fractions was theoretically calculated using the model of Ørskov and McDonald [1] and the application of the Fit curve macro NEWAY Excel developed by Chen [37). This point has been clarified in Lines 176-182.

Point 4: Line 177-179: Why did the gas was collected only 1 mL from incubation and transferred into 5 mL glass vacuum airtight tubes? And ? mL gas was injected into the GC? And why didn`t collected the gas samples at 48 and 72 hours for methane detection?

Response 4: Thanks a lot for this valuable comment. The gas was collected at five time points (3, 6, 9, 12, and 24 hours). Thus, representative samples for the five time points were collected by taking 1 mL from each time point and transferr the 5 mL into 5 mL glass vacuum airtight tubes. Then, 1 mL from the collected 5 mL was injected into the GC to be representative to all time points. This point has been described in detail in the method section as directed (Lines 183-186). We have collected the gas samples till the 24 hrs only as most equations used for calculating the gas production kinetics, partitioning factor, SCFA, and ME depend on the gas collected in 24 hrs. Hence, many earlier studies follow the same protocol and collected the gas until 24hr only (Makkar et al., 2004; Jiménez-Peralta et al., 2011).  

Makkar, Harinder PS. Recent advances in the in vitro gas method for evaluation of nutritional quality of feed resources. Assessing quality and safety of animal feeds, 2004, 160: 55. 

Jiménez-Peralta, F. S., Salem, A. Z. M., Mejia-Hernández, P., González-Ronquillo, M., Albarrán-Portillo, B., Rojo-Rubio, R., Tinoco-Jaramillo, J. L. (2011). Influence of individual and mixed extracts of two tree species on in vitro gas production kinetics of a high concentrate diet fed to growing lambs. Livestock Science136(2-3), 192-200.

Point 5: 3. Results; Table 4. How to calculate Main effect?

Response 5: Thanks for your comment. The “main effects” were changed to “Overall mean” and it has been claculated using the Mixed procedure of SAS (version 9.0, SAS Institute Inc., Cary, NC, USA) as clarifirf in the statistical section (Lines 216-217). Also, this point has been clarified in the table footnote as directed (Lines 310-311).

Point 6: Line 320. GPSF and GPNSF should be clarified when first used.

Response 6: Revised as directed (Lines 32-322).

Point 7: Table 6: TOMD or TDOM?

Response 7: Thanks a lot for this valuable observation. The correct is “TDOM”. It has been corrected.

Point 8: Table 4, Table 5, Table 6: In this study, cumulative gas production was measured at different incubation times, and parameters of gas production were measured at 72 h. And In table 6, PF, methane production, ammonia-nitrogen, and microbial protein were measured at 24 h of incubation time. Why were the times of measurements different?

Response 8: Because the equations used for calculating the PF and microbial protein depend on the gas collected in 24 hrs. The references used for these parameters have been mentioned in lines 211-214. However, the most estimated parameters were taken after 72 hr such as pH, ammonia-nitrogen, VFA, and feed degradability.
